# At Stability's Edge: How to adjust hyper-parameters to preserve minima selection in Asynchronous Training of Neural Networks?

**Niv Giladi**[1]*    **Mor Shpigel Nacson**[1]*    **Elad Hoffer**[1,2]    **Daniel Soudry**[1]

[1]Technion - Israel Institute of Technology, Haifa, Israel
[2]Habana-Labs, Caesarea, Israel
`{giladiniv, mor.shpigel, elad.hoffer, daniel.soudry}@gmail.com`

## Abstract

**Background:** Recent developments have made it possible to accelerate neural networks training significantly using large batch sizes and data parallelism. Training in an asynchronous fashion, where delay occurs, can make training even more scalable. However, asynchronous training has its pitfalls, mainly a degradation in generalization, even after convergence of the algorithm. This gap remains not well understood, as theoretical analysis so far mainly focused on the convergence rate of asynchronous methods.
**Contributions:** We examine asynchronous training from the perspective of dynamical stability. We find that the degree of delay interacts with the learning rate, to change the set of minima accessible by an asynchronous stochastic gradient descent algorithm. We derive closed-form rules on how the learning rate could be changed, while keeping the accessible set the same. Specifically, for high delay values, we find that the learning rate should be kept inversely proportional to the delay. We then extend this analysis to include momentum. We find momentum should be either turned off, or modified to improve training stability. We provide empirical experiments to validate our theoretical findings.

## 1 Introduction

Training deep neural networks (DNNs) requires large amounts of computational resources, often using many devices operating over days and weeks. Furthermore, with ever-increasing model size and available data, the amount of compute used was noted to increase exponentially over the past few years (Amodei & Hernandez, 2018). Fortunately, the operations made in DNN training are highly suitable for parallelization over many devices. Most commonly, the parallelization is done by "data-parallelism" in which the data is divided into separate batches which are distributed between different devices during training. By using many devices, with each device highly utilized, one could train on a large number of samples without paying in run-time. However, this training is done synchronously, with synchronization between the different devices done on every iteration (S-SGD). With the growth in the number of devices participating in the training, the synchronization becomes a prominent bottleneck.

To overcome the synchronization bottleneck, asynchronous training (A-SGD) has been proposed (Dean et al., 2012; Recht et al., 2011). The basic idea behind such training is that when a device finishes calculating its batch gradients, an update to the DNN parameters is immediately performed, without waiting for other devices. This approach is known as asynchronous centralized training with a parameter server (PS) that updates the parameters (Li, 2014). The main problem in such training is that, since the PS updates the parameters whenever a device communicates with him, the parameters that are being used in other devices calculation are no longer up-to-date. This phenomena is called *gradient staleness* or *delay* as we will refer to it, and it causes a deterioration in generalization performance when using A-SGD.

---

*Equal contribution

The generalization deterioration can be seen in Fig. 1. Unless the learning rate is significantly decreased, we can see a *generalization gap*: a deterioration in the generalization of A-SGD (with delay $\tau = 32$) from the baseline (S-SGD, $\tau = 0$) value *near the steady state* — i.e., after we are near full convergence, following many training epochs (2000). Due to the generalization gap demonstrated in the figure, A-SGD with a parameter server and large delays is not commonly used — despite it is relatively simple to implement and despite its vast potential to accelerate training. Therefore, our main goal in this work is to shed some theoretical light on the generalization gap problem and use it to improve A-SGD training.

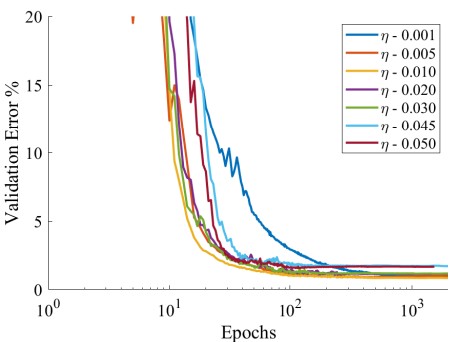 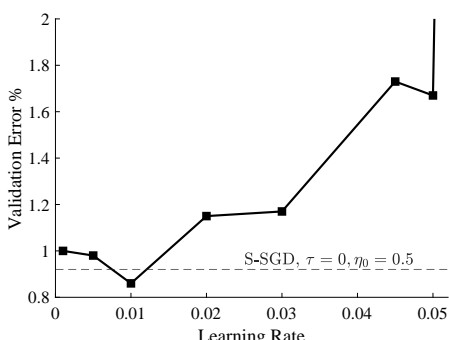

Figure 1: **Impact of learning rate and delay on generalization error.** Validation error with delay ($\tau = 32$) and different learning rates $\eta$. We observe that unless we decrease the learning rate (proportionally to $1/\tau$) there is a generalization gap from the equivalent large batch training: (left) training curve. (right) validation error as a function of learning rate after reaching steady state. Horizontal dashed line is the validation error acquired by the equivalent large batch training (S-SGD, $\tau = 0$). Learning rates larger then 0.05 did not converge with ($\tau = 32$) delay. CNN trained with MNIST. More details in section E.

Previous theoretical analysis of A-SGD (Liu et al., 2018; Lian et al., 2016; 2015; Dai et al., 2018; Dutta et al., 2018; Arjevani et al., 2018) focused on analyzing the convergence rate, i.e., the time it takes to reach steady state, and not on the properties of the obtained solution. In contrast, in our paper we focus on understanding how the delay affects the selection of the solution we converge to, and changes in this selection can impact generalization.

We tackle these questions from the perspective of dynamical stability. The dynamical stability approach was used in (Nar & Sastry, 2018; Wu et al., 2018), to study which minima are accessible under a specific choice of optimization algorithm and hyperparameters. In (Nar & Sastry, 2018), the authors analyzed gradient descent (GD) algorithm as a discrete-time nonlinear dynamical system. By analyzing the Lyapunov stability of this system for different minima, they showed a relation between the learning rate choice and the accessible minima set, i.e., the subset of the local optima that GD can converge to. In (Wu et al., 2018), the authors focused on stochastic gradient descent (SGD), defined a criterion to evaluate the stability of a specific minimum and used this criterion to show how the learning rate and batch-size play a role in SGD minima selection process.

Here, we use dynamical stability to analyze the dynamics of A-SGD. Using this approach, the main question we try to tackle is:

*How do learning rate, delay and momentum interact and affect the minima selection process?*

**Contributions**

We start by modelling A-SGD as a dynamical system with delay. By analyzing the stability properties of that system, we find:

- There exist an inverse linear relation between the delay of A-SGD, and the threshold learning rate i.e., the critical learning rate in which a minimum loses stability.
- This implies that in order to keep a specific minimum stability as we change the delay, we need to change the learning rate inversely proportional to the delay.
- Momentum has a crucial role in determining stability properties for A-SGD. Depending on the specific training algorithm, it can either hurt the stability or improve it. We show how to modify momentum to stabilize A-SGD training.

With our theoretical findings, we derive closed form rules when training asynchronously. We suggest to scale the learning rate inversely to the delay in order to reach a better generalization performance. In section 3 we provide experiments on popular classification tasks to support our theoretical analysis. It is interesting to contrast this simple linear relation with the more complicated picture observed for the scaling of the learning rate with the batch size. Specifically, it has been observed by Shallue et al. (2018) that the optimal learning rate scales differently with the batch size when changing the models and datasets, i.e., there is no general "rule of thumb" that applies in all cases.

A similar scaling rule for the learning rate was proposed before (Zhang et al., 2015), however, there it lacked a theoretical basis or a demonstration of its effectiveness with large effective batch sizes, i.e., the sum of all the minibatch sizes of all the workers, as we do here.

## 2 THEORETICAL ANALYSIS

### 2.1 PRELIMINARIES AND PROBLEM SETUP

Consider the problem of minimizing the empirical loss

$$f(\mathbf{x}) = \sum_{i=1}^{N} f_i(\mathbf{x}) \,, \tag{1}$$

where $N$ is the number of samples, $\mathbf{x} \in \mathbb{R}^d$ and $\forall i = 1, ..., N$: $f_i : \mathbb{R}^d \to \mathbb{R}$ is twice continuously differentiable function. The update rule for minimizing $f(\mathbf{x})$ in eq. 1 using asynchronous stochastic gradient descent (A-SGD) is given by

$$\mathbf{x}_{t+1} = \mathbf{x}_t - \eta \nabla f_{n(t-\tau)}(\mathbf{x}_{t-\tau}) \,, \tag{2}$$

where $\eta$ is the learning rate, $n(t)$ is a random selection process of a sample from $\{1, 2, \ldots, N\}$ at iteration $t$, and $\tau$ is some delay due the asynchronous nature of our training. For simplicity, we focus on a fixed delay $\tau$. The extension to stochastic delay is discussed in section 2.2.1.

Our main goal is understanding how different factors such as the gradient staleness $\tau$ and the learning rate $\eta$ affect the minima selection process in neural networks. In other words, we would like to understand how the different hyperparameters interact to divide the minima of the loss to two sets: those we can converge to and those we cannot converge to.

To analyze this we suppose we are in a vicinity of some minimum point $\mathbf{x}^*$, which implies $\nabla f(\mathbf{x}^*) = 0$. Given some values of $\eta$ and $\tau$, we ask whether or not we can converge to $\mathbf{x}^*$, using the theory of dynamical stability. As we shall see, if the learning rate $\eta$ or delay $\tau$ are too high, then this minimum loses stability, i.e., A-SGD cannot converge to $\mathbf{x}^*$, in expectation.

Specifically, we examine the expectation of the linearized dynamics around $\mathbf{x}^*$. We show in appendix A that, to ensure stability, it is sufficient that the following one-dimensional equation is stable:

$$x_{t+1} - x_t + \eta a x_{t-\tau} = 0 \,, \tag{3}$$

where we defined the Hessian $\mathbf{H} = \frac{1}{N} \sum_{i=1}^{N} \nabla^2 f_i(\mathbf{x}^*)$, the sharpness term $a = \lambda_{\max}(\mathbf{H})$ as the maximal eigenvalue of the Hessian, and $x_t$ is the projection of the expectation of the perturbation $\mathbf{x}_t - \mathbf{x}^*$ on the eigenvector which corresponds to the maximal eigenvalue $a$. Additionally, we show in appendix A that the characteristic equation of eq. 3 is

$$z^{\tau+1} - z^\tau + \eta a = 0 \,. \tag{4}$$

### 2.2 STABILITY ANALYSIS

For given $(\eta, \tau)$, we would like to determine if the minimum point $\mathbf{x}^*$ is asymptotically stable (Oppenheim, 1999) in expectation. This will enable us to divide the minima into two sets: those we might converge to since they are stable, and those we cannot possibly converge (except for a measure zero set of initialization) since they are unstable. We use the characteristic equation (eq. 4) to define the stability criterion:

**Definition 1 (Stability criterion)** *For given $(\eta, \tau)$, we say that the dynamics in eq. 3 are stable if all the roots of the corresponding characteristic equation (eq. 4) are inside the unit circle. This condition is equivalent to $x_t \to 0$ for any initialization.*

We would like to find conditions on when and how can we change the hyperparameters $(\eta, \tau)$ to maintain the same stability criterion. This, for example, will enable us to understand how to compensate for the delay introduced by asynchrony using the learning rate. In order to analyze stability we examine the characteristic equation in eq. 4 and ask: when does the maximal root of this polynomial have unit magnitude?

### 2.2.1 THE INTERACTION BETWEEN LEARNING RATE AND DELAY

Recall the characteristic equation

$$z^{\tau+1} - z^\tau + a\eta = 0 \,. \tag{5}$$

This polynomial has $\tau + 1$ roots. In order to ensure stability we require that the root with the maximal magnitude, i.e., the maximal root, will be inside the unit circle. We want to find the threshold learning rate, i.e., the learning rate in which the maximal root is exactly on the unit circle, so we are at the threshold of stability. In appendix B, we show that this requires that

$$a\eta = 2\sin\left(\frac{\pi}{4\tau + 2}\right) \,. \tag{6}$$

Using Taylor approximation we get

$$a\eta = 2\left(\frac{\pi}{4\tau + 2} + O\left(\frac{1}{\tau^3}\right)\right) \Rightarrow \frac{1}{a\eta} = \frac{1}{\pi}(2\tau + 1) + O\left(\frac{1}{\tau}\right) \,. \tag{7}$$

We observe numerically that the error between the exact solution (eq. 6) and the linear approximation (eq. 7) is smaller than $0.05$ for $\tau \geq 1$. Additionally, in the appendix Fig. 6 we demonstrate the high accuracy of the analytic approximation of the relation between $\frac{1}{a\eta}$ and $\tau$, compared to the numerical evaluation of the value of $\frac{1}{a\eta}$ for which the maximal root of eq. 5 is on the unit circle.

**Implications:** We can see that to maintain stability for a given minimum point (or a given sharpness value) the learning rate should be kept inversely proportional to the delay. This implies that given some minimum $\mathbf{x}^*$ for $\tau = 0$ and a corresponding learning rate $\eta_0$ for which this minimum is stable, we can evaluate the learning rate which will ensure this minimum remain stable for larger delay values. To do this, we first estimate $a$ using eq. 6: $a\eta_0 = 2\sin\left(\frac{\pi}{4\cdot 0 + 2}\right) = 2 \Rightarrow a = \frac{2}{\eta_0}$. Note that we do not use eq. 7 to evaluate $a$ since we are interested in $\tau = 0$, while the relation in eq. 7 is only a good approximation for $\tau \geq 1$. Next, given some delay $\tau$, we substitute $a$ into eq. 7 in order to evaluate a learning rate value for which the stability of all minima is maintained as in $\tau = 0$

$$\eta \approx \frac{\pi}{a(2\tau + 1)} = \frac{\pi\eta_0}{4(\tau + 0.5)} \,. \tag{8}$$

**Related empirical results**: In section 3.1 we show empirically the existence of a stability threshold for different values of delay and compare it with our theoretical threshold.

**Stochastic delay:** In appendix C we extend our theoretical analysis to derive the characteristic equation for A-SGD with stochastic delay, i.e., when $\tau$ is drawn from some discrete distribution. We analyze two cases:

- Discrete uniform distribution: $\tau \sim \text{unif}\{a, b\}$.
- Gaussian distribution discrete approximation: $\forall \mu, \forall k = 1, ..., 2\mu : \Pr(\tau = k) = \Pr(k - 0.5 \leq \mathbf{z} \leq k + 0.5)$ where $\mathbf{z} \sim \mathbb{N}(\mu, 1)$. In Zhang et al. (2015), the authors showed empirically a delay distribution that resembles to Gaussian distribution.

For both cases, we show numerically that the inverse relation between the learning rate $\eta$ and the expected delay $\mathbb{E}\tau$ still applies.

### 2.2.2 THE EFFECTS OF MOMENTUM ON STABILITY

The optimization step of A-SGD with momentum is defined as follows:

$$\mathbf{v}_{t+1} = m\mathbf{v}_t - \eta\,(1-m)\,\nabla f_{n(t-\tau)}(\mathbf{x}_{t-\tau})$$
$$\mathbf{x}_{t+1} = \mathbf{x}_t + \mathbf{v}_{t+1}\,, \tag{9}$$

where $\mathbf{x}_t$ are the weights, $\mathbf{v}_t$ and $\nabla f_{n(t)}(\mathbf{x}_t)$ are the velocity and gradients at time $t$, respectively, $m$ is the momentum parameter, $\eta$ is the step size and $\tau$ is the delay. The constant $(1-m)$ term which multiplies the gradients is called "dampening".

In appendix D we extend our theoretical analysis to derive the characteristic equation for A-SGD *with momentum*. In addition, in section D.2.1 we show numerically that an inverse relation between the learning rate $\eta$ and the delay $\tau$ also applies when using momentum. Particularly,

1. We need to keep the learning rate inversely proportional to the delay.
2. For a given delay, larger momentum values require smaller learning rate for stability.

Therefore, the range of stable learning rates decreases when the momentum parameter ('$m$') increases which can make tuning the learning rate difficult, especially with large delays. These results suggest that, to ensure stability when training with large delay, it is recommended to work without momentum.

**Relation to previous results:** The conclusion that when using A-SGD with high delay values it's recommended to turn off momentum is consistent with the results of Mitliagkas et al. (2017); Liu et al. (2018). In Liu et al. (2018), for streaming PCA, the authors showed it is necessary to reduce momentum in order to ensure convergence and acceleration through asynchrony. In Mitliagkas et al. (2017) the authors suggested that asynchrony adds implicit momentum to the training process and therefore it is necessary to reduce momentum when working asynchronously.

Next, we discuss a new method for introducing momentum into asynchronous training. This method enables asynchronous training where increasing the momentum parameter improves the stability.

**Shifted momentum** As discussed, training asynchronously is less stable when momentum is used. We propose to utilize momentum in an alternative method, called shifted momentum, for training asynchronously where the momentum benefits the training process and particularly its stability. We observe this method tends to improve the convergence stability compared to training without momentum (or with the original momentum), as is demonstrated in appendix Fig. 11. We suggest that instead of exchanging the gradients terms, as in eq. 9, we exchange the entire velocity term. This way, each worker calculates the velocity term based in its own gradients and momentum buffer. The formulation of this method is given in the following equation:

$$\mathbf{v}_{t+1} = m\mathbf{v}_{t-\tau} - \eta\,(1-m)\,\nabla f_{n(t-\tau)}(\mathbf{x}_{t-\tau})$$
$$\mathbf{x}_{t+1} = \mathbf{x}_t + \mathbf{v}_{t+1}$$

or

$$\mathbf{x}_{t+1} = \mathbf{x}_t + m\,(\mathbf{x}_{t-\tau} - \mathbf{x}_{t-\tau-1}) - \eta\,(1-m)\,\nabla f_{n(t-\tau)}(\mathbf{x}_{t-\tau}) \tag{10}$$

In appendix D we show that the characteristic equation of eq. 10 is

$$z^{\tau+2} - z^{\tau+1} + (\eta\,(1-m)\,a - m)\,z + m = 0\,, \tag{11}$$

where we use the sharpness term $a = \lambda_{\max}(\mathbf{H})$ as before.

We would like to characterize how the learning rate, delay and momentum affect minima stability. Thus, for each momentum value, we evaluate numerically the value of $\frac{1}{a\tau}$ for which the maximal root of eq. 11 is on the unit circle, i.e., when we are on the threshold of stability. In Fig. 4 we see that the inverse relation between the learning rate $\eta$ and the delay $\tau$ still applies when using shifted momentum. Moreover, we can see that in contrast to regular momentum, in shifted momentum, a larger momentum value $m$ enables us to work with larger step-size. Therefore, increasing the momentum *improves* the stability.

**Relation to previous results:** The analysis of delayed gradients and velocity might also shed light on the stability of recent asynchronous training approaches, where the workers communicating through weight averaging (Lian et al., 2017; Assran et al., 2019). This method is similar to ours in the sense that each worker holds its own momentum buffer. In parallel to us, (Hakimi et al., 2019) used a variation of shifted momentum with Nesterov momentum (denoted as Multi-ASGD), and demonstrated empirically its generalization benefits, but without a theoretical analysis of this benefit.

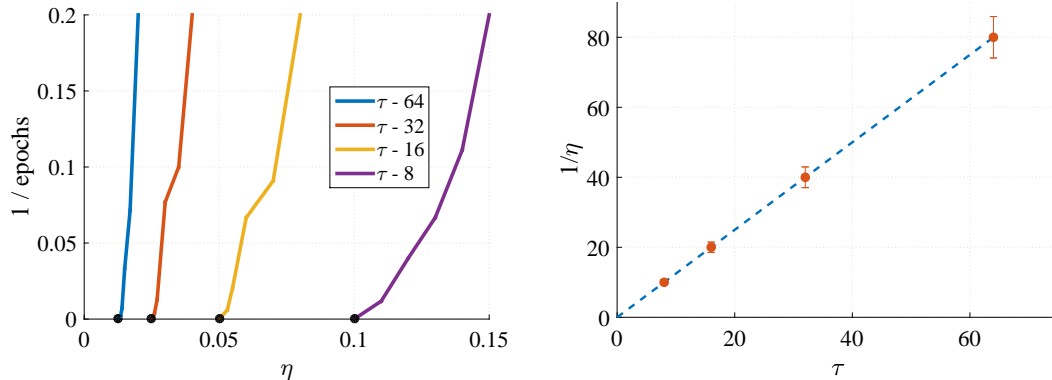

Figure 2: **Stability threshold is maintained when $\eta \propto 1/\tau$**: In the left figure we show the number of epochs it takes to diverge from a minimum as a function of the learning rate $\eta$. The black circles are the stability thresholds — below which, we do not escape the minimum. In the right figure for each value of delay $\tau$ we show $1/\eta$ where $\eta$ is the maximal learning rate in which we did not diverge. Due to sampling resolution, there might be up to 8% deviation from the maximal learning rate found. This deviation is represented in the error bars. VGG-11 trained with CIFAR10.

## 3 EXPERIMENTS

In this section, we provide experiments to support our theoretical findings. In the following subsections we: (1) demonstrate the relationship between hyperparameters and minima selection, and (2) show our findings can help improve asynchronous training. Details about the experiments and the implementation can be found in appendix E. [1]

### 3.1 HOW HYPERPARAMETERS AFFECT MINIMA SELECTION

In this section, we demonstrate how the delay and learning rate affect the accessible minima set. We train with a fixed learning rate.

**Minima stability.** To demonstrate how delay and learning rate change the stability of a minimum, we start from a model that converged to some minimum and train with different values of delay and learning rate to see if the algorithm leaves that minimum, i.e., the minimum becomes unstable. We do so by training a VGG-11 on CIFAR10 for 10,000 epochs – until we reach a steady state. This training is done without delay, momentum and weight decay. Next, introduce a delay $\tau$, change the learning rate, and continue to train the model. Fig. 2 shows the number of epochs it takes to leave the steady state for a given delay $\tau$ and learning rate $\eta$. We observe that for certain $(\tau, \eta)$ pairs, the algorithm stays in the minimum (below the black circle) while for others it leaves that minimum after some number of epochs (above the black circle). Importantly, we can see in the right panel in Fig. 2 that, as predicted by the theory, the inverse learning rate, $1/\eta$, where $\eta$ is the maximal learning rate in which we did not diverge, scales linearly with the delay $\tau$. Additional details are given in appendix F.

**Generalization with delay.** With delay, we need to adjust the hyperparameters in order to maintain the accessible minima set. However, even if a certain pair of $(\tau,\eta)$ changes the accessible set, it is possible that there are still accessible minima that generalize well. We perform experiments to investigate what type of generalization we can expect from training with different $(\tau,\eta)$ pairs. We examine this by training a VGG-11 on CIFAR10 with different $(\tau,\eta)$ pairs until we reach a steady state (around 6,000 epochs). We use plain SGD without momentum or weight decay. We compare the generalization performance achieved by each pair of learning rate and delay. We repeat the experiment for each pair four times and report mean and standard deviation values. The results are presented in Fig. 3. We can see a linear scaling between the delay values and the learning rate empirical stability threshold, i.e., the learning rate in which the error diverge. In addition, we see that the smallest error (with small variance) is obtained at a learning rate which is somewhat smaller then the learning rate at the empirical threshold of stability, as expected (LeCun et al., 2012).

---

[1]Code will be available at `https://github.com/paper-submissions/delay_stability`

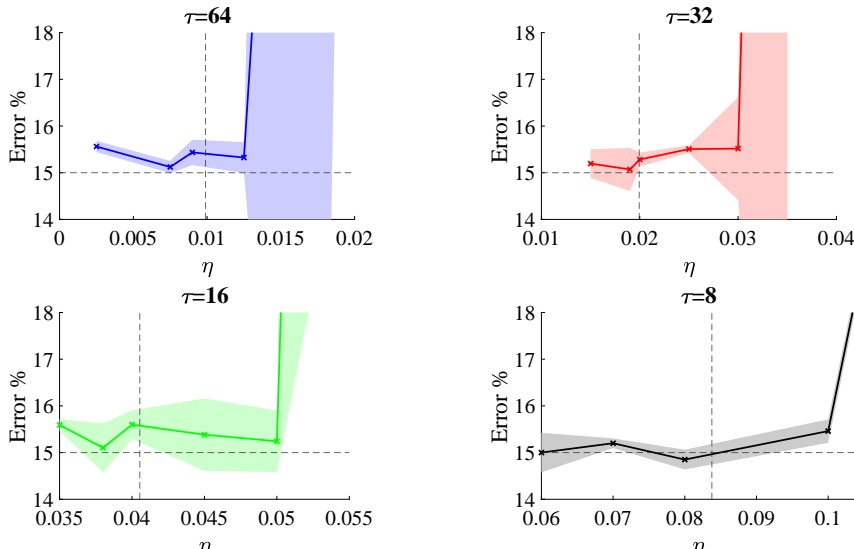

Figure 3: **Better generalization obtained near the stability threshold.** Validation error Vs. learning rate $\eta$ of VGG-11 trained with CIFAR10 for different delay values. Solid lines represent mean validation error. The margins represent one standard deviation. The vertical dashed line is the learning rate according to eq. 8. The horizontal dashed line is the accuracy obtained with $\tau=0$ (S-SGD).

**Shifted momentum stability.** In subsection 2.2.2 we introduced shifted momentum as a modification of the standard momentum update, in which increasing the momentum value ('$m$') improves stability. We validate the stability properties of this method by training a fully connected model with MNIST. The model has 3 layers, 1024 neurons in each layer with ReLU activations. Again, training is done synchronously with a large batch and no weight decay until it reaches a steady state. Then, after reaching a minimum, we continued to train the model, each time with a different triplet of $(\tau, \eta, m)$. We found the maximum learning rate for a given $m$ and $\tau$ for which training does not diverge, i.e, this maximum learning rate is at the edge of stability. Fig. 4 depicts the empirical results. The value of $a = \lambda_{\max}(\mathbf{H})$ is estimated at the minimum point using power iteration. Two important results emerge from the graph. First, in order to maintain the stability threshold, the learning rate has to be inversely proportional to the delay, as described in subsection 2.2. Second, with increased momentum values, training becomes more stable. Meaning, we can use larger learning rates with larger momentum value. This stands in contrast to the analysis of the original momentum: as we show numerically in appendix D, increasing the momentum value $m$ in the standard momentum algorithm decreases the stability and hence we need to decrease the learning rate as the momentum value grows larger.

## 3.2    IMPROVING ASYNCHRONOUS TRAINING

So far we focused on the steady state behavior of A-SGD, i.e. after many iterations. However, one of the prominent motivations for training asynchronously is to speed-up training. In this subsection, we examine whether our findings are also relevant for improving the accuracy of models trained asynchronously with little to no excess budget of epochs, compared to equivalent large batch regime. In large batch training, the learning rate is scaled as a function of the ratio between the large batch size and the small one. As Shallue et al. (2018) pointed out, we can expect a different learning rate scaling for different models/datasets. We start with a set of models and datasets that follow a square root scaling of the learning rate with the batch size (Hoffer et al., 2017). Recall that our inversely linear scaling rule (i.e., $\eta \propto 1/\tau$) applies to the large batch learning rate. We demonstrate the validity of our inversely linear learning rate scaling by comparing our method with one that does not account for the delay $\tau$. The results are presented in Table 1. We can see that using the small batch regime with delay (ASGD) does not converge at all. With our inversely linear learning rate scaling (+LR), there is a small gap from the equivalent large batch. If we also increase the budget of epochs by 30%, our optimization regime (+30%) generalizes better than the large batch (LB). Because A-SGD has better run-time performance compared to S-SGD (Dutta et al., 2018; Assran et al., 2019), such excess in epochs may still result in improved overall wall clock time (this depends on the hardware details).

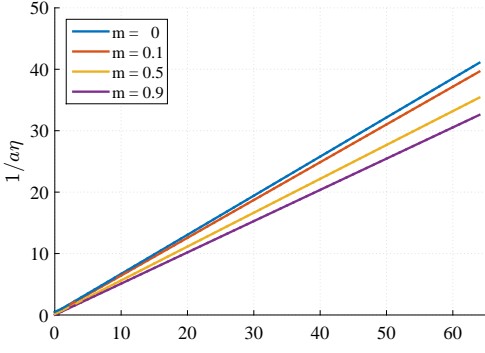 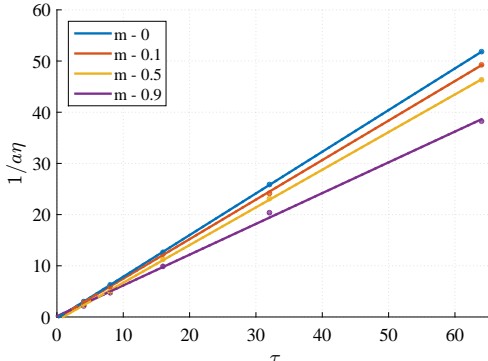

Figure 4: **Stability threshold is improved when in increasing the momentum parameter, when using shifted momentum**: (left) For different values of momentum $m$, we examine the relation between $\frac{1}{a\eta}$ and $\tau$ obtained by numerically evaluating the value of $\frac{1}{a\eta}$ for which the maximal roots of eq. 11 is on the unit circle. With shifted momentum, we maintain the stability threshold. (right) we evaluate empirically the relation depicted in the left panel by training, with shifted momentum, a fully connected model with MNIST. The markers are empirically evaluated, and the solid lines are qclinear fit to the markers.

Table 1: Validation accuracy results, SB/LB represent small and large batch respectively with $\tau = 0$. ASGD is asynchronous training with $\tau = 32$, +LR stands for ASGD with our linear scaling, and +30% is also with additional 30% epochs budget.

| Network | Dataset | SB | LB | ASGD | +LR | +30% |
|---|---|---|---|---|---|---|
| Resnet44 (He et al., 2016) | Cifar10 | 92.87% | 90.42% | 10.0% | 88.57% | 91.65% |
| VGG11 (Simonyan & Zisserman, 2014) | Cifar10 | 89.8% | 84.55% | 10.0% | 61.41% | 84.74% |
| C3 (Keskar et al., 2016) | Cifar100 | 60.0% | 57.89% | 1.0% | 54.02% | 58.94% |
| WResnet16-4 (Zagoruyko & Komodakis, 2016) | Cifar100 | 76.78% | 72.85% | 17.8% | 70.35% | 73.81% |

**ImageNet.** We next experiment with ResNet50 and ImageNet, which is a popular benchmark for large batch and asynchronous training because of its complexity and scale. Our focus is asynchronous centralized training with high delay. To the best of our knowledge, the generalization gap is not closed yet in this setting. This is an important setting because of its simplicity and its scalability potential. Optimization details can be found in appendix E.1

In Fig. 5, we compare three A-SGD optimizations trained with ImageNet. As can be seen in the figure, the generalization gap decreases from $4.33\%$ to $1.69\%$, just by turning off momentum. This corresponds with our analysis and previous results about the relation between momentum and delay. Shifted momentum similarly improves generalization, as can be seen in the right panel. This validates our analysis on the beneficial role the momentum parameter has, using shifted momentum. The learning rate is first scaled linearly with the total batch size of all the workers, as in Goyal et al. (2017). Our inverse linear scaling with delay cancels this increase, and we get a similar learning rate as in regular small batch training. In contrast, when we scaled the learning rate with $1/\sqrt{\tau}$, training did not converge at all.

## 4 DISCUSSION

**Motivation.** A fundamental problem with asynchronous training is that it seems to cause a degradation in generalization, even after training has converged to a steady state. Identifying the root of this problem and resolving it is essential in order to scale parallelism. When approaching this problem, we need to address two issues. First, what is the origin of this generalization gap? Second, how can we decrease this generalization gap?

**Conclusions.** Our work tackles both questions by studying the behaviour of A-SGD and the relation between the delay and the hyperparameters, e.g., the learning rate and momentum. We analyze A-SGD from the theoretical perspective of dynamical stability. We find that a linear connection between the inverse learning rate and delay is necessary in order to keep the set of accessible minima

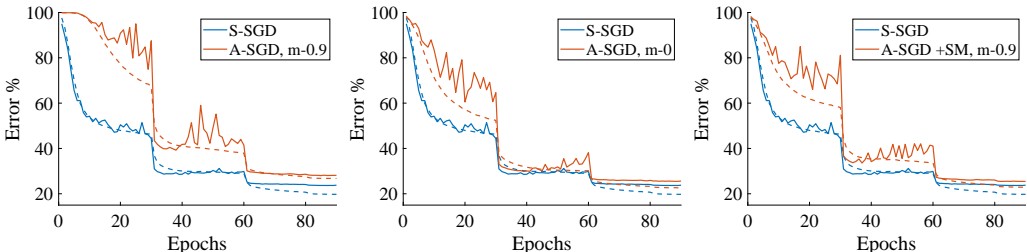

Figure 5: **ResNet50 ImageNet asynchronous training.** The baseline (in blue, S-SGD) is a large batch synchronous training, following Goyal et al. (2017), where $\eta \propto$ batch size. For A-SGD, based on our analysis, we additionally change $\eta \propto 1/\tau$ (in contrast, $\eta \propto 1/\sqrt{\tau}$ does not converge at all). Thus, in this case, we get the same learning rate as in a small batch regime. We compare three A-SGD options (in orange): (1) with standard momentum (2) with momentum value $m = 0$; and (3) with shifted momentum. S-SGD validation final error is 23.8%. A-SGD achieves 28.13% when training with standard momentum (left). The error drops to 25.49% when turning off momentum (middle). With shifted momentum, the error is similar, 25.4% (right). This matches the analysis in section 2.2.2. Solid lines are validation error and dashed lines are train error.

the same. In addition, we find how momentum affects the accessible minima set. Our analysis suggests that, when increasing delay, momentum degrades stability. Therefore, it should be turned off in order to maintain performance, as was previously observed empirically (Mitliagkas et al., 2017; Dai et al., 2018; Liu et al., 2018). However, we observe that momentum can have some benefits in improving convergence stability without delay. To maintain these benefits, we propose to modify momentum into a method called shifted momentum. We supply a theoretical analysis of A-SGD with shifted momentum and conclude that, in contrast to regular momentum, this method improves stability when delay is increased. In addition, we supply empirical evidence to support these findings. Lastly, we demonstrate, these findings can improve the generalization performance of distributed training algorithms and bring us one step closer to closing the generalization gap.

**Future Directions.** It is interesting to examine how the dynamical stability analysis holds during training. Specifically, what implications can we derive from the analysis, before we reach the steady state (e.g., a minimum of the loss), given a tight epochs budget. Our empirical findings in this case (Fig. 13 ) suggest that the optimal learning rate chosen for the steady state may remain optimal during the convergence process. This might be because the hyperparameters at the stability threshold are closely related to the hyperparameters that achieve the optimal convergence speed to that minimum. For example, in GD it easy to show that $\eta_{\text{threshold}} = 2\eta_{\text{optimal}}$. Therefore, both stability and convergence rate may be optimally maintained using similar scaling relations in the hyperparameters. This is an interesting question for future study.

Another interesting question for future study is to understand how the stability conditions change when some synchronization is introduced. Several practical A-SGD methods use synchronization to some extent, e.g. (Assran et al., 2019; Chen et al., 2016). For synchronization methods that reduce the delay in expectation, we can intuitively expect an improvement in dynamical stability, from our analysis of stochastic delay (appendix C). However, further analysis of the stability method for each method is required for an exact answer.

Answering these questions may lead to a more precise understanding of the trade-offs between S-SGD and the various A-SGD methods.

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

# Appendix

## A   STABILITY ANALYSIS FOR EQ. 2

In this section, we would like to analyze the stability of eq. 2. In particular, we would like to examine if a given minimum point $\mathbf{x}^*$ is stable. To analyze this we suppose we are in a vicinity of some minimum point $\mathbf{x}^*$, which implies $\nabla f(\mathbf{x}^*) = 0$, and ask whether or not we can converge to $\mathbf{x}^*$ using the theory of dynamical stability. Formally, consider the linearized dynamics around $\mathbf{x}^*$:

$$\mathbf{x}_{t+1} = \mathbf{x}_t - \eta\left(G(\mathbf{x}^*; n_{t-\tau}) + \nabla G\left(\mathbf{x}^*; n_{t-\tau}\right)\left(\mathbf{x}_{t-\tau} - \mathbf{x}^*\right)\right),$$

where we denoted $G(\mathbf{x}_t; n_t) = \nabla f_{n_t}(\mathbf{x}_t)$ and changed the notation of the time index in $n(t)$ to the subscript, i.e., $n_t$.

Examining the expectation of this equation we obtain

$$\mathbb{E}\mathbf{x}_{t+1} \overset{(1)}{=} \mathbb{E}\mathbf{x}_t - \eta\left(\frac{1}{N}\sum_{i=1}^{N}G(\mathbf{x}^*; i) + \frac{1}{N}\sum_{i=1}^{N}\nabla G\left(\mathbf{x}^*; i\right)\left(\mathbb{E}\mathbf{x}_{t-\tau} - \mathbf{x}^*\right)\right)$$

$$\overset{(2)}{=} \mathbb{E}\mathbf{x}_t - \eta\mathbf{H}\left(\mathbb{E}\mathbf{x}_{t-\tau} - \mathbf{x}^*\right),$$

where in (1) we used $\forall k = 1, ..., N : \Pr(n_t = k) = \frac{1}{N}$, and in (2) we used the fact that $\mathbf{x}^*$ is a minimum point and thus $\sum_{i=1}^{N} G(\mathbf{x}^*; i) = \nabla f(\mathbf{x}^*) = 0$ and denoted $\mathbf{H} = \frac{1}{N}\sum_{i=1}^{N}\nabla G(\mathbf{x}^*; i) = \frac{1}{N}\sum_{i=1}^{N}\nabla^2 f_i(\mathbf{x}^*)$.

Using $\mathbf{s}_t = \mathbb{E}\mathbf{x}_t - \mathbf{x}^*$ variable change we obtain

$$\mathbf{s}_{t+1} = \mathbf{s}_t - \eta\mathbf{H}\mathbf{s}_{t-\tau}. \tag{12}$$

From the Spectral Factorization Theorem we can write $\mathbf{H} = \mathbf{U}\mathbf{A}\mathbf{U}^\top$ where $\mathbf{U}$ is an orthogonal matrix and $\mathbf{A} = \mathrm{diag}(a_1, ..., a_n)$ is a diagonal matrix. Substituting this into eq. 12 we obtain:

$$\mathbf{s}_{t+1} = \mathbf{s}_t - \eta\mathbf{U}\mathbf{A}\mathbf{U}^\top\mathbf{s}_{t-\tau}.$$

Multiplying the last equation with $\mathbf{U}^\top$ from the left, denoting $\tilde{\mathbf{s}}_t = \mathbf{U}^\top\mathbf{s}_t$, and using the fact that $\mathbf{U}$ is an orthogonal matrix, i.e., $\mathbf{U}^\top\mathbf{U} = \mathbf{I}$ we get:

$$\tilde{\mathbf{s}}_{t+1} = \tilde{\mathbf{s}}_t - \eta\mathbf{A}\tilde{\mathbf{s}}_{t-\tau}.$$

Recall that $\mathbf{A} = \mathrm{diag}(a_1, ..., a_N)$ is a diagonal matrix. Therefore, analyzing the dynamics of the last equation is equivalent to analyzing the dynamics of the following $N$ one-dimensional equations:

$$\forall i = 1, ...N \ : \ \tilde{\mathbf{s}}_{t+1}(i) = \tilde{\mathbf{s}}_t(i) - \eta a_i\tilde{\mathbf{s}}_{t-\tau}(i),$$

where $\mathbf{s}_t(i)$ denotes the $i^{\text{th}}$ component of the vector $\mathbf{s}_t$.

In order to ensure stability of the $N$-dimensional dynamical system, it's sufficient to require that

$$\tilde{\mathbf{s}}_{t+1}(1) = \tilde{\mathbf{s}}_t(1) - \eta a_1\tilde{\mathbf{s}}_{t-\tau}(1), \tag{13}$$

where we assume without loss of generality $a_1 \geq a_2 \geq \cdots \geq a_N$. This is true since the dynamics for $i = 1$ is the first one that loses stability.

To simplify notations we define $x_t = \tilde{\mathbf{s}}_t(1)$ and the sharpness term $a = a_1 = \lambda_{\max}(\mathbf{H})$, the maximal singular value of $\mathbf{H}$. Using these notations, we can re-write eq. 13 as:

$$x_{t+1} = x_t - \eta a x_{t-\tau} \Leftrightarrow x_{t+1} - x_t + \eta a x_{t-\tau} = 0. \tag{14}$$

Since this is a linear time invariant difference equation, of order $\tau + 1$, its solution is, generically:

$$x_t = \sum_{i=1}^{\tau+1} c_i z_i^t,$$

where $c_i$ are determined by the initial conditions, and $z_i$ are the roots of the characteristic equation of eq. 14. To find this characteristic equation, we substitute $x_t = z^t$ into eq. 14, and obtain

$$z^{t+1} - z^t + \eta a z^{t-\tau} = 0 \Leftrightarrow z^{\tau+1} - z^\tau + \eta a = 0$$

Note that for $x_t \to 0$, it is required that $\forall i, |z_i| < 1$.

## B   Finding the condition for the maximal root of the characteristic equation (eq. 4) to be on the unit circle

To simplify the notation, we denote $\alpha = a\eta$. Recall the characteristic eq. (eq. 5)

$$z^{\tau+1} - z^\tau + \alpha = 0\,.$$

Since we are looking for solution on the unit circle we substitute $z = e^{i\theta}$ into the equation where $i$ is the unit imaginary number and $\theta$ is some angle. We obtain:

$$e^{i\theta(\tau+1)} - e^{i\theta\tau} + \alpha = 0\,.$$

Using $e^{i\theta} = \cos(\theta) + i\sin(\theta)$ we get

$$\cos\left(\theta(\tau+1)\right) + i\sin\left(\theta(\tau+1)\right) - \left(\cos\left(\theta\tau\right) + i\sin\left(\theta\tau\right)\right) + \alpha = 0 \Rightarrow$$

$$\begin{cases} \cos\left(\theta(\tau+1)\right) - \cos\left(\theta\tau\right) + \alpha = 0 \\ \sin\left(\theta(\tau+1)\right) - \sin\left(\theta\tau\right) = 0 \end{cases}$$

Using trigonometric product-sum identities we obtain

$$\begin{cases} -2\sin\left(\frac{\theta(2\tau+1)}{2}\right)\sin\left(\frac{\theta}{2}\right) + \alpha = 0 \\ 2\cos\left(\frac{\theta(2\tau+1)}{2}\right)\sin\left(\frac{\theta}{2}\right) = 0 \end{cases} \tag{15}$$

If $\sin\left(\frac{\theta}{2}\right) = 0$ then we get from the first equation $\alpha = 0$. This is a contradiction since we assume that $\alpha > 0$. Thus, $\sin\left(\frac{\theta}{2}\right) \neq 0$. This implies from the second equation in eq. 15 that

$$\cos\left(\frac{\theta(2\tau+1)}{2}\right) = 0 \Rightarrow \frac{\theta(2\tau+1)}{2} = \frac{\pi}{2} + \pi k\,, \text{ for } k = 0, 1, ..., 2\tau\,.$$

Substituting this result into the first equation in eq. 15 and using $\alpha > 0$ we obtain that the possible solutions must satisfy

$$\alpha = 2\sin\left(\frac{(1+4k)\pi}{4\tau+2}\right)\,, \text{ for } k = 0, 1, ..., \tau\,.$$

Also, since $\sin(x) = \sin(\pi - x)$ we can eliminate symmetric solution and get

$$\alpha = 2\sin\left(\frac{(1+4k)\pi}{4\tau+2}\right)\,, \text{ for } k = 0, 1, ..., \lfloor\frac{\tau+1}{2}\rfloor\,.$$

Note that for $k = 0, 1, ..., \lfloor\frac{\tau+1}{2}\rfloor$: $0 < \frac{(1+4k)\pi}{4\tau+2} < \frac{\pi}{2}$ and thus $\sin\left(\frac{(1+4k)\pi}{4\tau+2}\right) > 0$ is monotonically increasing with $k$. Since the magnitude of the characteristic equation increases with $\alpha$ and we are interested in the maximal root we are looking for the minimal $\alpha$ value. This value corresponds to $k = 0$. Thus, the maximal root satisfies:

$$\alpha = \sin\left(\frac{\pi}{4\tau+2}\right)\,.$$

## C   Stability Analysis for eq. 2 with stochastic delay

We assume that $\tau(t)$ is drawn from some distribution. In this section, we would like to analyze the stability of eq. 2. In particular, we would like to examine if a given minimum point $\mathbf{x}^*$ is stable. To analyze this we suppose we are in a vicinity of some minimum point $\mathbf{x}^*$, which implies $\nabla f(\mathbf{x}^*) = 0$, and ask whether or not we can converge to $\mathbf{x}^*$ using the theory of dynamical stability. Formally, consider the linearized dynamics around $\mathbf{x}^*$:

$$\mathbf{x}_{t+1} = \mathbf{x}_t - \eta\left(G(\mathbf{x}^*; n_{t-\tau(t)}) + \nabla G\left(\mathbf{x}^*; n_{t-\tau(t)}\right)\left(\mathbf{x}_{t-\tau(t)} - \mathbf{x}^*\right)\right)\,,$$

where we denoted $G(\mathbf{x}_t; n_t) = \nabla f_{n_t}(\mathbf{x}_t)$ and changed the notation of the time index in $n(t)$ to the subscript, i.e., $n_t$.

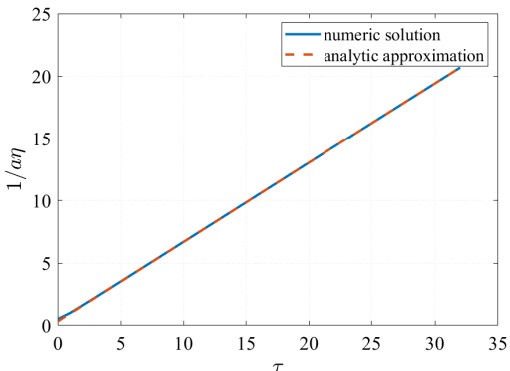

Figure 6: **It is necessary to keep the learning rate inversely proportional to the delay to maintain stability.** In blue we see the relation between $\frac{1}{a\eta}$ and $\tau$ obtained by numerically evaluating the value of $\frac{1}{a\eta}$ for which the maximal root of eq. 5 is on the unit circle. In orange we see the analytic approximation $\frac{1}{a\eta} = \frac{1}{\pi}(2\tau + 1)$. We can see that in order to maintain stability for a given minimum point, as the delay $\tau$ increases we need to decrease the learning rate $\eta$ to keep $\eta$ inversely proportional to $\tau$. In appendix (Fig. 10) we show that with momentum, we also get similar linear relation, only with a higher slope.

Examining the expectation of this equation we obtain

$$\mathbb{E}\mathbf{x}_{t+1} \overset{(1)}{=} \mathbb{E}\mathbf{x}_t - \eta \left( \frac{1}{N}\sum_{i=1}^{N} G(\mathbf{x}^*; i) + \frac{1}{N}\sum_{i=1}^{N} \nabla G(\mathbf{x}^*; i)\left( \sum_k \mathbb{E}\mathbf{x}_{t-k}\Pr(\tau(t) = k) - \mathbf{x}^* \right) \right)$$

$$\overset{(2)}{=} \mathbb{E}\mathbf{x}_t - \eta\mathbf{H}\sum_k \Pr(\tau(t) = k)(\mathbb{E}\mathbf{x}_{t-k} - \mathbf{x}^*) ,$$

where in (1) we used $\forall t$ and $\forall k = 1, ..., N : \Pr(n_t = k) = \frac{1}{N}$, and in (2) we used the fact that $\mathbf{x}^*$ is a minimum point and thus $\sum_{i=1}^{N} G(\mathbf{x}^*; i) = \nabla f(\mathbf{x}^*) = 0$ and denoted $\mathbf{H} = \frac{1}{N}\sum_{i=1}^{N} \nabla G(\mathbf{x}^*; i) = \frac{1}{N}\sum_{i=1}^{N} \nabla^2 f_i(\mathbf{x}^*)$ .

Using $\mathbf{s}_t = \mathbb{E}\mathbf{x}_t - \mathbf{x}^*$ variable change we obtain

$$\mathbf{s}_{t+1} = \mathbf{s}_t - \eta\mathbf{H}\sum_k \Pr(\tau(t) = k)\mathbf{s}_{t-k} . \tag{16}$$

From the Spectral Factorization Theorem we can write $\mathbf{H} = \mathbf{U}\mathbf{A}\mathbf{U}^\top$ where $\mathbf{U}$ is an orthogonal matrix and $\mathbf{A} = \text{diag}(a_1, ..., a_n)$ is a diagonal matrix. Substituting this into eq. 16 we obtain:

$$\mathbf{s}_{t+1} = \mathbf{s}_t - \eta\mathbf{U}\mathbf{A}\sum_k \Pr(\tau(t) = k)\mathbf{U}^\top\mathbf{s}_{t-k} .$$

Multiplying the last equation with $\mathbf{U}^\top$ from the left, denoting $\tilde{\mathbf{s}}_t = \mathbf{U}^\top\mathbf{s}_t$, and using the fact that $\mathbf{U}$ is an orthogonal matrix, i.e., $\mathbf{U}^\top\mathbf{U} = \mathbf{I}$ we get:

$$\tilde{\mathbf{s}}_{t+1} = \tilde{\mathbf{s}}_t - \eta\mathbf{A}\sum_k \Pr(\tau(t) = k)\tilde{\mathbf{s}}_{t-k} .$$

Recall that $\mathbf{A} = \text{diag}(a_1, ..., a_N)$ is a diagonal matrix. Therefore, analyzing the dynamics of the last equation is equivalent to analyzing the dynamics of the following $N$ one-dimensional equations:

$$\forall i = 1, ...N \ : \ \tilde{\mathbf{s}}_{t+1}(i) = \tilde{\mathbf{s}}_t(i) - \eta a_i \sum_k \Pr(\tau(t) = k)\tilde{\mathbf{s}}_{t-k}(i) ,$$

where $\mathbf{s}_t(i)$ denotes the $i^{\text{th}}$ component of the vector $\mathbf{s}_t$.

In order to ensure stability of the $N$-dimensional dynamical system, it's sufficient to require that

$$\tilde{\mathbf{s}}_{t+1}(1) = \tilde{\mathbf{s}}_t(1) - \eta a_1 \sum_k \Pr(\tau(t) = k)\tilde{\mathbf{s}}_{t-k}(1), \tag{17}$$

where we assume without loss of generality $a_1 \geq a_2 \geq \cdots \geq a_N$. This is true since the dynamics for $i = 1$ is the first one that loses stability.

To simplify notations we define $x_t = \tilde{\mathbf{s}}_t(1)$ and the sharpness term $a = a_1 = \lambda_{\max}(\mathbf{H})$, the maximal singular value of $\mathbf{H}$. Using these notations, we can re-write eq. 13 as:

$$x_{t+1} = x_t - \eta a \sum_k \Pr(\tau(t) = k)x_{t-k} \Leftrightarrow x_{t+1} - x_t + \eta a \sum_k \Pr(\tau(t) = k)x_{t-k} = 0. \tag{18}$$

The characteristic equation of eq. 18 is

$$z - 1 + \eta a \sum_k \Pr(\tau(t) = k)z^{-k} = 0 \tag{19}$$

## C.1 STABILIY ANALYSIS

We would like to find conditions on when and how can we change the hyperparameters $(\eta, \mathbb{E}\tau)$ to maintain the same stability criterion (definition 1). In order to analyze stability we examine the characteristic equation in eq. 18 and ask: when does the maximal root of this polynomial have unit magnitude?

1. $\tau \sim \text{unif}\{a, b\}$:

$$z - 1 + \eta a \sum_{k=a}^{b} \frac{1}{n}z^{-k} = 0 \tag{20}$$

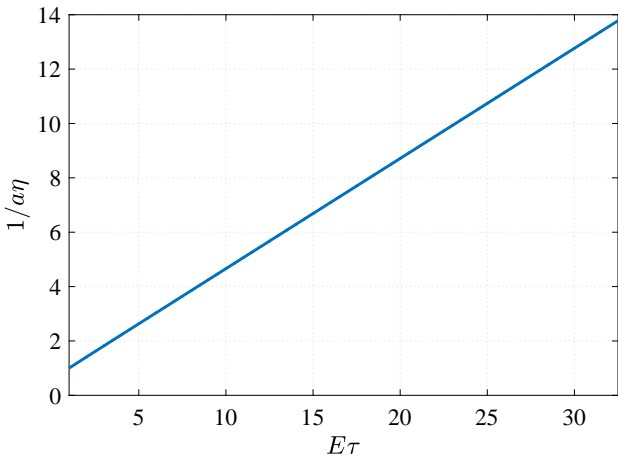

Figure 7: $\tau \sim \text{unif}\{a, b\}$ where $a = 1$ and $b$ is an integer in the range $[1, 64]$. In this case, $\mathbb{E}\tau = \frac{a+b}{2}$. We see the relation between $\frac{1}{a\eta}$ and $\mathbb{E}\tau$ obtained by numerically evaluating the value of $\frac{1}{a\eta}$ for which the maximal root of eq. 20 is on the unit circle. We can see that in order to maintain stability for a given minimum point, as the delay expectation $\mathbb{E}\tau$ increases we need to decrease the learning rate $\eta$ to keep $\eta$ inversely proportional to $\tau$.

2. Gaussian distribution discrete approximation: $\forall \mu, \forall k = 1, ..., 2\mu : \Pr(\tau = k) = \Pr(k - 0.5 \leq \mathbf{z} \leq k + 0.5)$ where $\mathbf{z} \sim \bar{\mathbb{N}}(\mu, 1)$:

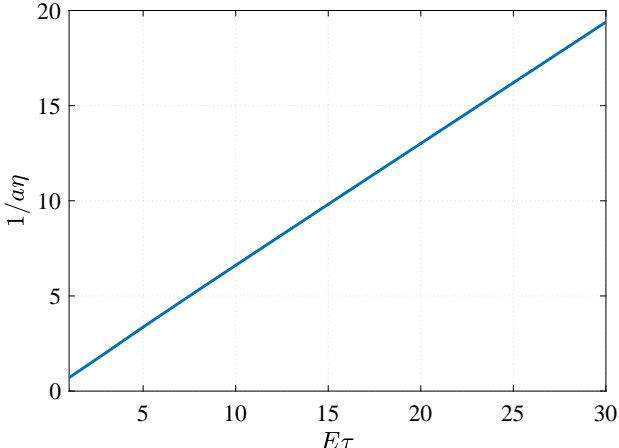

Figure 8: $\forall\mu, \forall k = 1, ..., 2\mu : \Pr(\tau = k) = \Pr(k - 0.5 \leq \mathbf{z} \leq k + 0.5)$ where $\mathbf{z} \sim \mathbb{N}(\mu, 1)$ and $\mu$ is an integer in the range $[1, 30]$. In this case, $\mathbb{E}\tau = \mu$. We see the relation between $\frac{1}{a\eta}$ and $\mathbb{E}\tau$ obtained by numerically evaluating the value of $\frac{1}{a\eta}$ for which the maximal root of eq. 19 is on the unit circle. We can see that in order to maintain stability for a given minimum point, as the delay expectation $\mathbb{E}\tau$ increases we need to decrease the learning rate $\eta$ to keep $\eta$ inversely proportional to $\tau$.

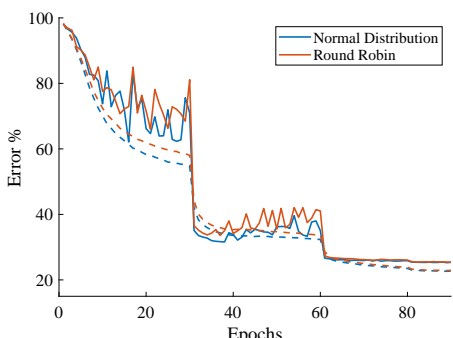

Figure 9: **Gaussian distribution of delay.** Comparison of ResNet50 trained asynchronously on ImageNet with two types of delay distributions: constant i.e., round robin (orange) and discrete Gaussian distribution (blue). Constant distribution reaches $25.4\%$ validation error while Gaussian distriution reaches a similar error of $25.3\%$.

## D    THEORETICAL ANALYSIS WITH MOMENTUM

In this section, we repeat the steps of the analysis in section 2 for A-SGD **with momentum**.

### D.1    PRELIMINARIES AND PROBLEM SETUP

Consider the problem of minimizing the empirical loss

$$f(\mathbf{x}) = \sum_{i=1}^{N} f_i(\mathbf{x}), \tag{21}$$

where $N$ is the number of samples, $\mathbf{x} \in \mathbb{R}^d$ and $\forall i = 1, ..., N$: $f_i : \mathbb{R}^d \to \mathbb{R}$ is twice continuously differentiable function. The update rule for minimizing $f(\mathbf{x})$ in eq. 21 using asynchronous stochastic gradient descent with momentum (A-MSGD) is given by

$$\mathbf{v}_{t+1} = m\mathbf{v}_t - \eta(1-m) G(\mathbf{x}_{t-\tau}; n_{t-\tau})$$
$$\mathbf{x}_{t+1} = \mathbf{x}_t + \mathbf{v}_{t+1}$$

or

$$\mathbf{x}_{t+1} = \mathbf{x}_t + m(\mathbf{x}_t - \mathbf{x}_{t-1}) - \eta(1-m) G(\mathbf{x}_{t-\tau}; n_{t-\tau}), \tag{22}$$

where $m$ is the momentum, $\eta$ is the learning rate, $G(\mathbf{x}_t; n_t) = \nabla f_{n_t}(\mathbf{x}_t)$, $n_t$ is a random selection process of a sample from $\{1, 2, \ldots, N\}$ at iteration $t$, and $\tau$ is some delay due the asynchronous nature of our training.

Our main goal is understanding how different factors such as the gradient staleness $\tau$, and hyperparameters such as the momentum $m$ and learning rate $\eta$ affect the minima selection process in neural networks. In other words, we would like to understand how the different hyperparameters interact to divide the global minima of our network to two sets: those we can converge to and those we cannot converge to.

To analyze this we suppose we are in a vicinity of a minimum point $\mathbf{x}^*$, which implies $\nabla f(\mathbf{x}^*) = 0$, and ask whether or not we can converge to $\mathbf{x}^*$ using the theory of dynamical stability. Formally, consider the linearized dynamics around some minimum point $\mathbf{x}^*$:

$$\mathbf{x}_{t+1} = \mathbf{x}_t + m(\mathbf{x}_t - \mathbf{x}_{t-1}) - \eta(1-m)(G(\mathbf{x}^*; n_{t-\tau}) + \nabla G(\mathbf{x}^*; n_{t-\tau})(\mathbf{x}_{t-\tau} - \mathbf{x}^*)).$$

Examining the first moment of this equation we obtain

$$\mathbb{E}\mathbf{x}_{t+1} = \mathbb{E}\mathbf{x}_t + m(\mathbb{E}\mathbf{x}_t - \mathbb{E}\mathbf{x}_{t-1}) - \eta(1-m)\left(\frac{1}{N}\sum_{i=1}^{N} G(\mathbf{x}^*; i) + \frac{1}{N}\sum_{i=1}^{N} \nabla G(\mathbf{x}^*; i)(\mathbb{E}\mathbf{x}_{t-\tau} - \mathbf{x}^*)\right)$$

$$= \mathbb{E}\mathbf{x}_t + m(\mathbb{E}\mathbf{x}_t - \mathbb{E}\mathbf{x}_{t-1}) - \eta(1-m)\mathbf{H}(\mathbb{E}\mathbf{x}_{t-\tau} - \mathbf{x}^*),$$

where $\frac{1}{N}\sum_{i=1}^{N} G(\mathbf{x}^*; i) = \frac{1}{N}\nabla f(\mathbf{x}^*) = 0$ since $\mathbf{x}^*$ is a minimum point and $\mathbf{H} = \frac{1}{N}\sum_{i=1}^{N} \nabla G(\mathbf{x}^*; i) = \frac{1}{N}\sum_{i=1}^{N} \nabla^2 f_i(\mathbf{x}^*)$.

Using $\mathbf{s}_t = \mathbb{E}\mathbf{x}_t - \mathbf{x}^*$ variable change we obtain

$$\mathbf{s}_{t+1} = \mathbf{s}_t + m(\mathbf{s}_t - \mathbf{s}_{t-1}) - \eta(1-m)\mathbf{H}\mathbf{s}_{t-\tau}. \tag{23}$$

From the Spectral Factorization Theorem we can write $\mathbf{H} = \mathbf{U}\mathbf{D}\mathbf{U}^\top$ where $\mathbf{U}$ is an orthogonal matrix and $\mathbf{A} = \mathrm{diag}(a_1, ..., a_n)$ is a diagonal matrix. Substituting this into eq. 23 we obtain:

$$\mathbf{s}_{t+1} = \mathbf{s}_t + m(\mathbf{s}_t - \mathbf{s}_{t-1}) - \eta(1-m)\mathbf{U}\mathbf{A}\mathbf{U}^\top\mathbf{s}_{t-\tau}.$$

Multiplying the last equation with $\mathbf{U}^\top$ from the left, denoting $\tilde{\mathbf{s}}_t = \mathbf{U}^\top\mathbf{s}_t$, and using the fact that $\mathbf{U}$ is an orthogonal matrix, i.e., $\mathbf{U}^\top\mathbf{U} = \mathbf{I}$ we get:

$$\tilde{\mathbf{s}}_{t+1} = \tilde{\mathbf{s}}_t + m(\tilde{\mathbf{s}}_t - \tilde{\mathbf{s}}_{t-1}) - \eta(1-m)\mathbf{A}\tilde{\mathbf{s}}_{t-\tau}.$$

Recall that $\mathbf{A} = \text{diag}(a_1, ..., a_N)$ is a diagonal matrix. Therefore, analyzing the dynamics of the last equation is equivalent to analyzing the dynamics of the following $N$ one-dimensional equations:

$$\forall i = 1, ...N \; : \; \tilde{\mathbf{s}}_{t+1}(i) = \tilde{\mathbf{s}}_t(i) + m\left(\tilde{\mathbf{s}}_t(i) - \tilde{\mathbf{s}}_{t-1}(i)\right) - \eta\left(1 - m\right)a_i\tilde{\mathbf{s}}_{t-\tau}(i)\,,$$

where $\mathbf{s}_t(i)$ denotes the $i^{\text{th}}$ component of the vector $\mathbf{s}_t$.

In order to ensure stability of the $N$-dimensional dynamical system, it's sufficient to require that

$$\tilde{\mathbf{s}}_{t+1}(1) = \tilde{\mathbf{s}}_t(1) + m\left(\tilde{\mathbf{s}}_t(1) - \tilde{\mathbf{s}}_{t-1}(1)\right) - \eta\left(1 - m\right)a_1\tilde{\mathbf{s}}_{t-\tau}(1)\,, \tag{24}$$

where we assume without loss of generality $a_1 \geq a_2 \geq \cdots \geq a_N$. This is true since the dynamics for $a_1$ is the first one that loses stability.

To simplify notations we define $x_t = \tilde{\mathbf{s}}_t(1)$ and the sharpness term $a = a_1 = \lambda_{\max}\left(\mathbf{H}\right)$, the maximal singular value of $\mathbf{H}$. Using these notations, we can re-write eq. 24 as:

$$x_{t+1} = x_t + m\left(x_t - x_{t-1}\right) - \eta\left(1 - m\right)ax_{t-\tau} \Leftrightarrow$$
$$x_{t+1} - \left(m + 1\right)x_t + mx_{t-1} + \eta\left(1 - m\right)ax_{t-\tau} = 0\,. \tag{25}$$

The characteristic equation of this difference equation is

$$z^{\tau+1} - \left(1 + m\right)z^{\tau} + mz^{\tau-1} + \eta\left(1 - m\right)a = 0\,. \tag{26}$$

## D.2 STABILITY ANALYSIS

We would like to find conditions on when and how can we change the hyperparameters $(m, \eta, \tau)$ to maintain the same stability criterion (definition 1). In order to analyze stability we examine the characteristic equation in eq. 26 and ask: when does the maximal root of this polynomial have unit magnitude?

### D.2.1 THE INTERACTION BETWEEN LEARNING RATE, DELAY, AND MOMENTUM

For the general case and different values of momentum, we evaluate numerically the value of $\frac{1}{a\tau}$ for which the maximal root of the general characteristic equation (eq. 26) is on the unit circle, i.e., when we are on the threshold of stability. We show the results in Fig. 10. We observe that there is still an inverse relation between the learning rate and delay when using momentum. Specifically, we observe that:

1. We need to decrease the learning rate as the delay increases.

2. The lines becomes more steep for larger values of momentum. This implies that, for a given delay, maintaining stability for larger values of momentum requires smaller learning rate. Therefore, it is easier to maintain stability with $m = 0$.

## D.3 SHIFTED MOMENTUM

In this section, we repeat the steps of the analysis in section D.1 for A-SGD with **shifted** momentum. The update rule for minimizing $f(\mathbf{x})$ in eq. 21 using asynchronous stochastic gradient descent with shifted momentum is given by

$$\mathbf{v}_{t+1} = m\mathbf{v}_{t-\tau} - \eta\left(1 - m\right)\nabla f_{n(t-\tau)}(\mathbf{x}_{t-\tau})$$
$$\mathbf{x}_{t+1} = \mathbf{x}_t + \mathbf{v}_{t+1}$$

or

$$\mathbf{x}_{t+1} = \mathbf{x}_t + m\mathbf{v}_{t-\tau} - \eta\left(1 - m\right)\nabla f_{n(t-\tau)}(\mathbf{x}_{t-\tau})$$
$$= \mathbf{x}_t + m\left(\mathbf{x}_{t-\tau} - \mathbf{x}_{t-\tau-1}\right) - \eta\left(1 - m\right)\nabla f_{n(t-\tau)}(\mathbf{x}_{t-\tau})$$

Following the same steps as in section D.1 we obtain the following one-dimensional equation:

$$x_{t+1} = x_t + m\left(x_{t-\tau} - x_{t-\tau-1}\right) - \eta\left(1 - m\right)ax_{t-\tau} \Leftrightarrow$$
$$x_{t+1} - x_t + (\eta(1 - m)a - m)x_{t-\tau} + mx_{t-\tau-1} = 0\,.$$

The last equation characteristic equation is:

$$z^{\tau+2} - z^{\tau+1} + (\eta(1 - m)a - m)z + m = 0\,.$$

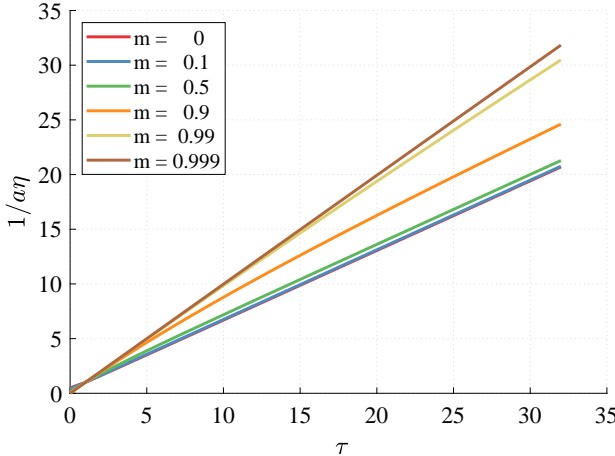

Figure 10: **Using (standard) momentum, it is still necessary to keep the learning rate inversely proportional to the delay to maintain stability.** For different values of momentum $m$, we examine the relation between $\frac{1}{a\eta}$ and $\tau$ obtained by numerically evaluating the value of $\frac{1}{a\eta}$ for which the maximal root of eq. 25 is on the unit circle. Similar to the results with $m = 0$, we can see that maintaining stability requires to keep the learning rate $\eta$ inversely proportional to the delay $\tau$.

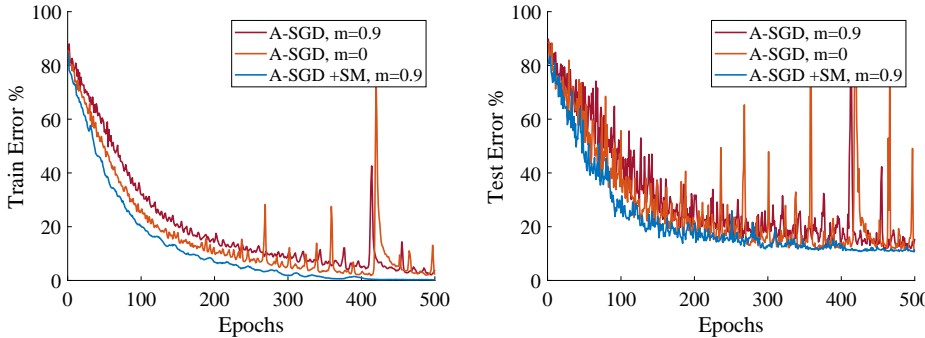

Figure 11: **Shifted momentum improves stability.** ResNet44 trained with CIFAR10 with same hyperparameters and three training algorithms: A-SGD with momentum (red), A-SGD without momentum (Orange) and A-SGD with shifted momentum (Blue). We observed "spikes" in the training error that appear when training with large batch size or with delay for large number of epochs, without decreasing the learning rate. While momentum negates these "spikes" in large batch training, Shifted momentum improves the convergence stability of the model and negates these "spikes" when training with delay.

## E    EXPERIMENTS DETAILS

We experimented with a set of popular classification tasks: MNIST (Lecun et al., 1998), CIFAR10, CIFAR100 (Krizhevsky, 2009) and ImageNet (Deng et al., 2009). In order to incorporate delay in the experiments, we keep replicas of the model parameters and perform SGD step with different replica at a time according to a round robin scheduling. The implementation is done with PyTorch framework. Section C in the appendix shows that constant delay, i.e., round robin, acts similarly in terms of stability to several other delay distributions. In addition, Zhang et al. (2015) shows empirically with realistic distribution that constant delay distribution is a good model for delay distribution. Fig. 9 shows empirically that round robin scheduling works in a similar way to discrete Gaussian distribtion.

Although stochastic gradient descent with momentum was originally introduced with the dampening term, many drop the latter when using momentum SGD. This is true for most DNN frameworks like Caffe, PyTorch, and TensorFlow where the dampening term is set to zero by default. As Shallue et al.

(2018); Yan et al. (2018) pointed out, without dampening, the momentum scales the learning rate so that the effective learning rate $\eta_{eff}$ is equal to $\eta_{eff} = \frac{\eta}{1-m}$ after sufficiently many iterations $T$ (i.e., when $T \to \infty$). Because we alter the momentum value in our experiments, we use dampening and scale the learning rate accordingly to keep the effective learning rate the same as without dampening.

### E.1 IMAGENET OPTIMIZATION DETAILS

Our baseline is Goyal et al. (2017) large batch training. They trained ResNet50 for 90 epochs with a large batch of 8192 samples and reached the same accuracy level of 23.8% as the small batch training. They used linear scaling with the batch size (scaling the learning rate by the large batch size divided by the baseline batch size) and a warm-up of the learning rate at the first 5 epochs. We use 32 workers, each with a minibatch size of 256.

## F MINIMA STABILITY EXPERIMENTS

In this section, we provide additional details about the minima stability experiment presented in section 3. As discussed in section 3, we are interested to examine for which pairs of $(\tau, \eta)$ the minima stability remains the same.

In Fig. 12 we show the validation error of such $(\tau, \eta)$ pairs as a function of epochs. We note that these graphs are the same experiment as in Fig. 2. As can be seen, for larger learning rates, it takes less epochs to leave the minimum. It is interesting to see that after leaving the minimum, the A-SGD algorithm converges again to a minimum with generalization as good as the baseline (at $\tau = 0$). This suggests that the minima selection process of A-SGD is affected by the whole optimization path. In other words, suppose we start the optimization from a minimum with good generalization (since it was selected using optimization with $\tau = 0$), and then it becomes unstable due to a change in the values of $(\eta, \tau)$, as in this experiment. These results in Fig. 12 suggest we typically converge to a stable minimum with similar generalization properties, possibly nearby the original minimum. In contrast, if we start to train from scratch using the same $(\eta, \tau)$ pair which lost stability in our experiment, we typically get a generalization gap (as observed in our experiments), which suggests the optimization path might have taken a very different path from the start, leading to other regions with worse generalization than the original minimum.

## G ADDITIONAL EXPERIMENTS

We examine what generalization performance we can expect during training, before reaching a minimum. We do so by using the same experiment in Fig. 1. We sample the validation error of each learning rate at different epochs. Results are presented in Fig. 13. Using our suggested learning rate modification (dividing the baseline large batch learning rate by the delay) gives the near optimal learning rate that can be seen in the figure.

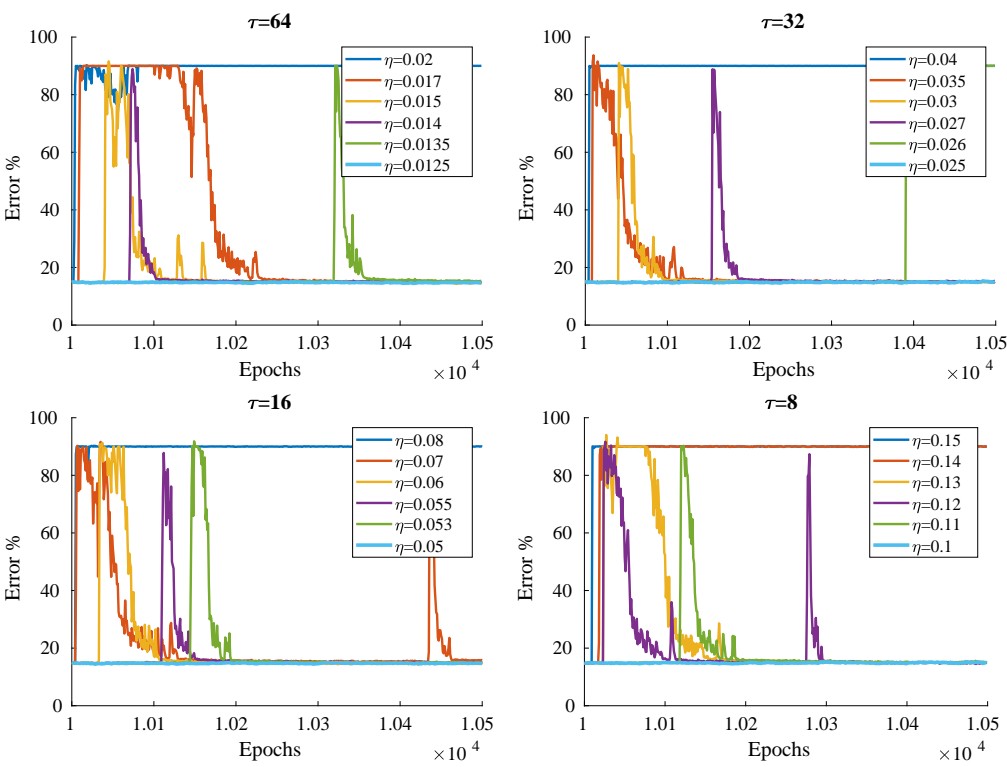

Figure 12: **Learning rates larger than $0.8/\tau$ diverge from the minimum.** We show the validation error Vs. epochs for different delay and learning rate values. The baseline learning rate used is $0.8$ according to large batch training Hoffer et al. (2017). The minimum learning rate for each $\tau$ is the stability threshold.

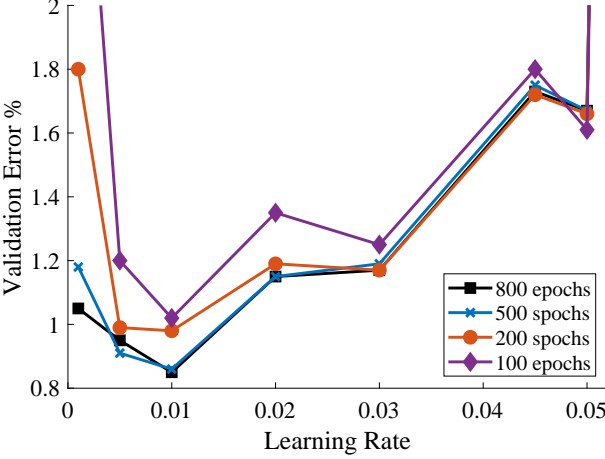

Figure 13: **Impact of learning rate and delay on generalization error during training.** Validation error with delay ($\tau = 32$) as a function of learning rate at different epochs. The optimal learning rate of $\eta = 0.01$ chosen found in the scan (which matches our proposed learning rate scaling), stays the same throughout the training. This might suggest why it's beneficial to use the optimal hyperparameters for the minimum, even when the algorithm hasn't converged to the steady state yet. CNN trained with MNIST, same settings as in Fig 1.

