# OpenReview forum: "At Stability's Edge: How to Adjust Hyperparameters to Preserve Minima Selection in Asynchronous Training of Neural Networks?"
_ICLR.cc/2020/Conference — Accept (Spotlight)_

### Official Review · AnonReviewer3 · 2019-10-22
**Official Blind Review #3**

**Rating:** 8

**Review:**

The authors model A-SGD as a dynamical system, where parameters are updated with delayed gradients. The authors analyze the stability of this system, and they first derive that the learning rate must scale linearly with the inverse of the delay around a minimum to remain stable. Using a similar analysis they show that the standard way of incorporating momentum into A-SGD requires small learning rates for high momentum values, and they propose "shifted momentum," which allows for stability under higher momentum values. Experimentally, the authors show that around minima the learning rate needed to retain stability scales linearly with the inverse of the delay, that there appears to be an analogous threshold when training models from scratch, that shifted momentum allows for higher momentum values, and finally that on several datasets A-SGD with an appropriate learning rate is able to generalize at least as well as large batch synchronous training.

This is a nice paper with a large number of interesting theoretical and experimental results, and I believe it should be accepted. I think there are some largely presentational issues that should be addressed, however:

- I think the authors should attempt to make a stronger case for the practical implications of their analysis: in particular, in the most practical setting (where we don't have a minimum obtained from synchronous training), what does the provided analysis allow us to do? Part of this might involve being more explicit about the results in Table 1: what exactly was the procedure for selecting the learning rates? Is it meaningfully different than just lowering the learning rate?

- Equation (3) is rather obscure without the Appendix, especially since unbolded x hasn't been introduced anywhere. I think the authors should try to convey more of what's going on in this equation in the main text.

Minor: 'looses' should be 'loses' throughout, and it might be good to include a conclusion section.

**Experience Assessment:**

I have read many papers in this area.

**Review Assessment: Checking Correctness Of Derivations And Theory:**

I assessed the sensibility of the derivations and theory.

**Review Assessment: Checking Correctness Of Experiments:**

I assessed the sensibility of the experiments.

**Review Assessment: Thoroughness In Paper Reading:**

I read the paper at least twice and used my best judgement in assessing the paper.

---

> ### Author Response · Authors · 2019-11-11
> **Reply to Reviewer #3**
>
> We thank the reviewer for the helpful feedback. We added a conclusions section as suggested. Below we address the questions the reviewer raised.
>
> (1a) "I think the authors should attempt to make a stronger case for the practical implications of their analysis: in particular, in the most practical setting (where we don't have a minimum obtained from synchronous training), what does the provided analysis allow us to do?"
>
> Our analysis is true for steady state, i.e., at the proximity of a minimum. As mentioned correctly, in practical cases, the training doesn't always end in a minimum. However, we observe that the training modifications we derive from our analysis help improve stability during training, even before reaching a minimum. We added empirical evidence that support this claim in appendix section G. In this section, we used the experiment in Fig. 1 and sampled the validation error in different epochs during training (see Figure 13). As can be seen, for learning rate = 0.01, which is the optimal learning for the steady state (and matches our proposed modification of the learning rate), the validation error stays near optimal throughout training.
>
> (1b) "Part of this might involve being more explicit about the results in Table 1: what exactly was the procedure for selecting the learning rates? Is it meaningfully different than just lowering the learning rate?"
>
> Our procedure for determining the A-SGD learning rate is to take the learning rate used for large batch training and divide it by the delay. In Table 1, the learning rate we used for the large batch is as suggested in [1]: the baseline (small batch) learning rate, multiplied by the square root of the ratio between the sizes of the small batch and the large batch.
>
> (2) "Equation (3) is rather obscure without the Appendix, especially since unbolded x hasn't been introduced anywhere. I think the authors should try to convey more of what's going on in this equation in the main text."
>
> We added more details on this equation in the main text. In particular, below equation (3) we added the definition of unbolded x: "$x_t$ is the projection of the expectation of the perturbation of $\mathbf{x}_t - \mathbf{x^{*}}$ on the eigenvector which corresponds to the maximal eigenvalue $a$".
>
> (3a) "Minor: 'looses' should be 'loses' throughout"
>
> We fixed this typo through the paper.
>
> (3b) "it might be good to include a conclusion section."
>
> We added a discussion section, which includes conclusions, and a discussion of implications (such as 1a above).
>
> [1] - Hoffer, E., Hubara, I.,  Soudry, D. (2017). Train longer, generalize better: closing the generalization gap in large batch training of neural networks.

---

### Official Review · AnonReviewer1 · 2019-10-24
**Official Blind Review #1**

**Rating:** 6

**Review:**

The authors introduce a theoretical model for delayed gradients in asynchronous training. It is a very nice model and solving the corresponding differential equation allows to study its stability. Authors derive stability bounds for pure SGD (learning rate needs to decrease with delay) and for SGD with momentum, where they introduce a nice momentum formulation that improves stability. These are nice insights and good results and they are validated by experiments. More experiments and practical analysis would be welcome though. Some example questions: would introducing some sychronization help? Is the lower learning rate hurting training speed when measures as wall-clock time to accuracy?

I am very grateful for the authors' response. It would still be good to see more experiments, but I hope this paper gets accepted.

**Experience Assessment:**

I have read many papers in this area.

**Review Assessment: Checking Correctness Of Derivations And Theory:**

I assessed the sensibility of the derivations and theory.

**Review Assessment: Checking Correctness Of Experiments:**

I carefully checked the experiments.

**Review Assessment: Thoroughness In Paper Reading:**

I read the paper at least twice and used my best judgement in assessing the paper.

---

> ### Author Response · Authors · 2019-11-11
> **Reply to Reviewer #1**
>
> We thank the reviewer for the positive and helpful feedback which allowed to improve the paper: we added a discussion section with two paragraphs elaborating on the Reviewer's questions, and how they lead into interesting directions for future research. We address the reviewer questions below:
>
> (1) "Would introducing some sychronization help?"
>
> There are several methods to incorporate some synchronization, e.g. [1,2].  We agree it would be an interesting research direction to investigate how our stability analysis changes for such synchronization methods. Generally, based on our analysis for asynchronous methods with stochastic delay (Appendix C), we expect that,  synchronization methods that reduce the delay in expectation will also help stability - enabling the use of larger learning rates. However, an exact answer would require choosing a specific method and calculating the stability threshold.
>
> (2) "Is the lower learning rate hurting training speed when measures as wall-clock time to accuracy?"
>
> In our experiments we did not find a degradation in convergence speed when using the optimal learning rate scaling for the steady state (naturally, the situation might change in other datasets or models). For example, in Figure 1 right, we see that the learning rate at steady state (after 2000 epochs) which achieves optimal generalization is 0.01. The same learning rate achieves optimal convergence, as can be seen in Figure 1 left: its validation error curve is almost always lower then all other learning rates. To see this more clearly, we also added an additional figure to the paper appendix G (Fig. 13) . In this new figure, we use the experiment introduced in Fig. 1 and show the validation error during training sampled at different epochs. As can be seen, with learning rate 0.01 (which matches the proposed modification of the learning rate), the validation error stays near optimal through training, i.e. at the different epochs sampled training we observe that the lower learning rate (chosen according to our suggested modification) achieves smaller validation error compared with higher or lower learning rates.
>
> Note that, although this is not wall-clock measuring, as there is no degradation in convergence speed in terms of iterations (or epochs), this implies that using the proposed learning rate in A-SGD will be beneficial in terms of run time performance as well - in comparison to other learning rates. If, instead, we compare A-SGD (with the proposed learning rate) vs. S-SGD (synchronous training), then, when measuring performance with wall-clock time, one should consider the system hardware, its heterogeneously, the network bandwidth and etc. There are settings in which training asynchronously will greatly benefit the training time compared to synchronous training. In such settings, using lower learning rate might improve wall-clock time to reach accuracy compared to synchronous training, as suggested by our results in section 3.2.
>
> [1] - Assran, M., Loizou, N., Ballas, N., Rabbat, M. (2018). Stochastic Gradient Push for Distributed Deep Learning.
>
> [2] - Chen, J., Pan, X., Monga, R., Bengio, S. (2016). Revisiting Distributed Synchronous SGD.

---

### Official Review · AnonReviewer4 · 2019-11-11
**Official Blind Review #4**

**Rating:** 8

**Review:**

This paper studies how asynchrony affects model training by investigating dynamic stability of minimum points that A-SGD can access. They point out that not all local minimum points are accessible, and asynchrony can affect which minimum points can be accessed, and thus helps to explain why models trained by A-SGD have higher generalization gap. The authors also propose shifted-momentum that utilize momentum for asynchronous training.

Overall, this paper provides nice insights and thorough theoretical analysis. Experiments are carefully designed to validate their results. I think this paper is well written and its novelty is significant.

Strength:
- Theoretical formulation and analysis in this paper is nice and elegant.
- Provide theoretical insights of A-SGD with momentum, which is important.
- Experiments of minima selection are carefully designed. I like the idea to observe trajectories ``leaving minimum''.

Some quick questions:
- In Fig. 3, we can clearly see a threshold of \eta. I notice that when \tau=16 the fluctuation is more significant than other three cases. Can you explain why this appears?
- In Sec. 3.1, do you consider any kind of learning rate scheduling to change learning rate over epochs, like you did in Sec. 3.2?
- It would be great to evaluate on more tasks, as it has been shown that some may be more robust than others (Dai et al., 2019).

Wei Dai, Yi Zhou, Nanqing Dong, Hao Zhang, and Eric Xing. Toward Understanding the Impact of Staleness in Distributed Machine Learning. In Proc. International Conference on Learning Representations (ICLR), 2019.

**Experience Assessment:**

I have read many papers in this area.

**Review Assessment: Checking Correctness Of Derivations And Theory:**

I assessed the sensibility of the derivations and theory.

**Review Assessment: Checking Correctness Of Experiments:**

I carefully checked the experiments.

**Review Assessment: Thoroughness In Paper Reading:**

I read the paper thoroughly.

---

> ### Author Response · Authors · 2019-11-13
> **Reply to Reviewer #4**
>
> We thank the reviewer for the positive and helpful feedback. Below we address the questions the reviewer raised.
>
> (1) "In Fig. 3, we can clearly see a threshold of $\eta$. I notice that when $\tau=16$ the fluctuation is more significant than other three cases. Can you explain why this appears?"
>
> Indeed, the standard deviation is somewhat larger when $\tau=16$. We suspect it is a finite-sample effect, and will run more repetitions to verify (it will take more than a few days to check thoroughly, but these results will be ready for the camera ready version).
>
> (2) "In Sec. 3.1, do you consider any kind of learning rate scheduling to change learning rate over epochs, like you did in Sec. 3.2?"
>
> The theoretical analysis in Section 2 focused on a fixed learning rate for simplicity. Therefore, in Section 3.1 which goal was to support our theoretical findings with empirical evidence, we chose to also focus on a fixed learning rate regime. Particularly, we investigated the interaction between the delay, a fixed learning rate, and momentum and how this interaction affects stability and generalization.
>
> It is interesting to explore the relation between the scheduling of the learning rate and the generalization from the perspective of dynamical stability. To do this, we need to consider different methods of practical learning rate scheduling regimes and analyze each scheduling method separately.
>
> (3) "It would be great to evaluate on more tasks, as it has been shown that some may be more robust than others (Dai et al., 2019)."
>
> Our findings of the relation between the momentum and asynchrony align with (Dai et al., 2019). In (Dai et al., 2019), the authors demonstrate that momentum based algorithms, e.g. Adam and RMSProp, are more sensitive to staleness. It is intriguing to analyze such algorithms in the asynchronous setting from the perspective of dynamical stability. We will aim to examine the behaviour of other tasks until the camera ready version (again, it will take us more than a few days to check it thoroughly). Thank you for the suggestion.

---

### Decision · Program_Chairs · 2019-12-19

**Decision:**

Accept (Spotlight)

**Comment:**

The paper considers the problem of training neural networks asynchronously, and the gap in generalization due to different local minima being accessible with different delays. The authors derive a theoretical model for the delayed gradients, which provide prescriptions for setting the learning rate and momentum.

All reviewers agreed that this a nice paper with valuable theoretical and empirical contributions.